# Changes in Physical Fitness Parameters in a Portuguese Sample of Adolescents during the COVID-19 Pandemic: A One-Year Longitudinal Study

**DOI:** 10.3390/ijerph20043422

**Published:** 2023-02-15

**Authors:** Cátia Silva, Catarina Vilas, Beatriz Pereira, Pedro Rosário, Sonia Fuentes, Paula Magalhães

**Affiliations:** 1Department of Applied Psychology, School of Psychology, University of Minho, 4710-052 Braga, Portugal; 2Facultad de Educación y Ciencias Sociales, Universidad Central de Chile, Santiago 530-598, Chile

**Keywords:** physical activity, school-aged sample, coronavirus disease 2019, longitudinal design

## Abstract

Due to a worldwide response to the COVID-19 pandemic, many changes were imposed on individuals’ daily lives, including those related to the physical activity of children and adolescents. The present study aims to comprehend the impact of early COVID-19 pandemic restrictions on Portuguese adolescents’ physical fitness parameters during two school years. A total of 640 students from the 5th to the 12th grades participated in the longitudinal study. Data on body composition, aerobic fitness, speed, agility, lower and upper body strength, and flexibility were collected at three moments: 1. before the COVID-19 pandemic (December 2019); 2. after the COVID-19 lockdown when the schools reopened delivering in-person classes (October 2020), and 3. two months after the in-person classes started (December 2020). To analyze the overall changes between the three moments and between two age groups, we conducted repeated measure ANOVAs. The main findings indicate that participants’ body composition (i.e., waist circumference) and aerobic fitness (i.e., maximal oxygen uptake) deteriorated after the first lockdown but improved two months after the in-person classes started. However, the same did not happen to neuromuscular fitness (i.e., horizontal Jumps and Sit and Reach). These findings suggest that the COVID-19 lockdown may have negatively impacted adolescents’ physical fitness, particularly older adolescents. Altogether, data reinforce the importance of in-person classes and school context in promoting adolescents’ physical health.

## 1. Introduction

Physical fitness—a set of attributes related to a person’s ability to perform physical and daily activities—is an important health marker and a strong predictor of future health [1,2]. Generally, physical fitness is influenced by a combination of genetic and environmental factors, for instance, a healthy diet and moderate-vigorous physical activity [3]. Recent literature has found associations between physical fitness, cardiovascular and skeletal health, muscular strength, and mental well-being. In addition, high levels of physical fitness during childhood and adolescence are likely to decrease cardiometabolic risk factors and health problems from an early age, positively affecting their future health [1].

During the last decades, influenced by changes in our lifestyle (e.g., increased use of technologies and changes in travel patterns), significant reductions in physical activity and changes in body composition were identified worldwide [2,4]. Moreover, a declining trend in physical fitness has been extensively reported, especially in cardiorespiratory fitness, flexibility, strength, and endurance, predicting severe health problems [1]. 

With the Coronavirus disease 2019 (COVID-19) pandemic outbreak and subsequent preventive measures, the declining trend of physical fitness has been aggravated for the youth population. In fact, due to a worldwide response to the COVID-19 pandemic, many changes (e.g., home confinement restrictions, school closure, and closure of extracurricular sports practices and recreational activities) were imposed on individuals’ daily lives. These changes altered family routines (e.g., inability to engage in sports) and affected peoples’ health-related behaviors [5,6,7], including children and adolescents’ physical activity, which may have increased obesity rates [8,9,10,11]. Recent findings have shown that youth’s physical fitness has been negatively affected by the pandemic constraints [12,13,14]. For instance, Tsoukos and Bogdanis [13] compared a group of adolescents (both females and males) who experienced COVID-19 lockdowns (2020–2021) against other pre-pandemic cohorts (2016–2017 and 2018–2019) and found that the former reported a higher Body Mass Index (BMI) and lower performance on several objectively measured fitness parameters.

Another recent study by Rúa-Alonso et al. [12] with two-independent samples of children and adolescents indicated changes in body composition (e.g., increase in BMI and waist circumference), particularly in boys, and a reduction in muscular fitness, especially in girls, during the pandemic years. However, no relevant changes in cardiorespiratory fitness in these samples were observed. Consistent with these findings, Lopez-Bueno et al. [14] found no significant changes in cardiorespiratory fitness (i.e., maximal oxygen uptake estimated by 20 m shuttle run test) in a sample of adolescents (12–14 years old) during COVID-19 lockdown. In contrast, Basterfield et al. [11] found that children’s BMI and cardiorespiratory fitness (i.e., 20 m shuttle run test performance) were more susceptible to change adversely over one year during lockdowns than were muscular fitness parameters (e.g., Handgrip Strength and Standing Broad Jump). Finally, a study with pre- and post-pandemic data reported that participants from 8 to 14 years old increased their BMI and reduced cardiovascular endurance (i.e., cardiorespiratory fitness) and muscle strength (i.e., Push-ups and Sit-ups) during the 2019–2020 school year, particularly older participants [15].

Literature provides growing evidence reporting physical fitness changes in children and adolescents during the COVID-19 pandemic. However, the results of these studies are mixed and need to be clearer [11,15]. Additionally, to the best of our knowledge, there is still little longitudinal data reporting the potential impact of pandemic lockdowns and other restrain measures (e.g., school closure) on people’s physical fitness parameters (e.g., speed, flexibility, aerobic fitness) [11,14,15,16]. Importantly, the longitudinal nature of these data is paramount to exploring how time spent in lockdown and after lockdown can influence adolescents’ physical fitness patterns over time. For this reason, more studies are needed. Ultimately, data on adolescents’ physical fitness patterns over time are limited-an age group living in a critical transition period between childhood and adulthood characterized by profound physiological changes throughout adolescence [14,17]. Considering early and middle adolescence may particularly help develop a deeper understanding of the differences in physical fitness during this period. For instance, adolescents in early adolescence are growing rapidly compared to middle adolescence (e.g., a stage where physical changes for females may be nearly complete) [3,17].

In sum, exploring the changes in physical fitness levels in early and middle adolescence during the COVID-19 pandemic and lockdown periods [15] and after-lockdown periods adds to the current corpus of research and is likely to help educators in their future practices (e.g., in monitoring and promoting physical activity and fitness in students during similar situations, for instance, deliver online classes; help parents engage in physical activity with their children). Thus, the present study aims to comprehend the impact of early COVID-19 pandemic restrictions on adolescents’ physical fitness parameters (e.g., body composition, aerobic fitness, speed, agility, lower and upper body strength, and flexibility). Data were gathered at three moments across two school years (2019/2020 and 2020/2021): Time 1 (T1)—December 2019, before the COVID-19 pandemic; Time 2 (T2)—October 2020 (i.e., after 10 months), after the COVID-19 lockdown when schools reopened for in-person classes; and Time 3 (T3)—December 2020, two months after the in-person classes started (i.e., 12 months after the first assessment—T1). Overall, we hypothesized that physical fitness parameters changed negatively after Portugal’s first lockdown due to the COVID-19 pandemic (i.e., from T1 to T2). However, these parameters were expected to improve as adolescents returned to school and their routines (i.e., from T2 to T3). Finally, we expected older participants to show less improvement in physical fitness parameters over time than their younger counterparts.

## 2. Materials and Methods

### 2.1. Sample and Setting

The sample comprised 640 participants between 10 and 18 years of age (*M* = 12.30; *SD* = 1.68), 305 (47.7%) of whom were girls. Following Allen and Waterman [17], the sample was divided into an early adolescent subgroup (*n* = 497; 77.7%), comprising participants aged between 10 and 13 years, and a middle adolescents’ subgroup (*n* = 143; 22.3%), comprising participants aged between 14 and 18 years. Moreover, from this sample, 575 adolescents were also assessed in T2 and 590 in T3, with just 7.8% dropouts over time (Table 1).

The data for this study was part of a large research project (CEICSH 032/2019) and was collected between 2019 and 2020. A convenience sample of students from grades 5 to 12 from two public schools in the north of Portugal was included. Of the total sample, 315 (49.2%) participants were from medium-high socio-economic family status, while 325 (50.8%) were from a low to extremely low socio-economic family status. Before this investigation, consent to conduct the study in a school setting was obtained from the Portuguese Ministry of Education and the University of Minho Ethics Committee for Research in Social and Human Sciences (CEICSH). In addition, written informed consent and assent were obtained from the parents/caregivers and the adolescents, respectively. Subsequently, sociodemographic information (e.g., sex), anthropometric measurements, and physical fitness parameters were recorded during school hours at three different moments (December 2019, October 2020, and December 2020) during two school years (2019/2020—2020/2021). The measurements were taken by the schools’ physical education (PE) teachers, who were previously trained to collect data following the protocols for each test. Individual codes were assigned to each participant to guarantee the confidentiality and anonymity of the data. The physical fitness assessment timeline can be found in Figure 1, embedded in the COVID-19 regulations that were implemented over time in schools.

### 2.2. Material and Measurements

#### 2.2.1. Sociodemographic Information 

Participants’ demographic information was collected regarding sex, age, education level (elementary, secondary, or higher), and family income (i.e., participants’ school social action level helped determine the family income).

#### 2.2.2. Physical Fitness

To assess and monitor the overall physical fitness of children and adolescents, eight physical fitness tests from the FITescola^®^ battery were implemented. FITescola is a digital platform that monitors fitness in youth aged 10 to 18 years attending Portuguese public schools. This program includes (i) anthropometric measurements (e.g., weight), (ii) aerobic cardiorespiratory fitness tests (e.g., shuttle run), and (iii) neuromuscular fitness tests (e.g., agility, speed). The physical fitness tests manual and videos, including procedures and protocols, are available from the FITescola^®^ online platform (http://fitescola.dge.mec.pt/HomeTestes.aspx (accessed on 28 November 2022)). 

Overall, good reliability has been reported for the tests comprising the FITescola^®^ battery in the general youth population [3]. All these measurements were assessed in the school context, during participants’ PE classes, on different weekdays. All tests were performed and supervised by trained PE teachers. Each one of these components of the FITescola battery is described as follows.

##### Anthropometric Measures and Body Composition

The anthropometric assessment included measuring height, weight, and waist circumference (WC) to assess body composition. All participants were weighed using an electronic scale. Then, their weight was measured using a stadiometer and their WC was measured with a measuring tape to test adiposity in the abdomen area of the participants. Subsequently, BMI for age and gender was calculated (BMI = Weight (kg)/Height^2^ (m). Finally, BMI was converted to Health Fitness Zone continuum (HFZc) scores, which reflects the relative difference of the participants’ BMI from the established age- and gender-health standards recommended by FITescola. These scores were calculated to identify changes over time [18]. For example, the formula used to calculate HFZc for BMI was:$$\mathrm{HFZc}=\frac{\left(\mathrm{BMI}-\mathrm{HFZ}\;\mathrm{BMI}\;\mathrm{threshold}\right)}{\mathrm{HFZ}\;\mathrm{BMI}\;\mathrm{threshold}}\;\times \;100$$

The HFZc score is reported as the percentage above or below the recommended HFZ BMI threshold for good health. For example, an HFZc score of +5% indicates that a participant’s BMI score is 5% above the HFZ BMI standard, whereas a score of −5% indicates that the participant’s BMI is within the HFZ BMI threshold by a 5% margin [18]. 

As suggested by Saint-Maurice et al. [18], this measure was specifically designed to overcome the limitations associated with BMI z-scores in capturing individual changes in BMI over time among youth who are overweight or have obesity.

##### Aerobic Fitness

Aerobic fitness, also known as cardiorespiratory fitness, is an important dimension of health-related fitness [19]. It reflects the overall capacity of the cardiovascular and respiratory systems [20]. A good aerobic fitness capacity during childhood and adolescence has been associated with a lower risk of cardiometabolic diseases, obesity, diabetes, and other health problems throughout the lifecycle [20]. In the FITescola program, the aerobic capacity was operationally defined as the estimated maximal oxygen uptake (VO2max). To assess this capacity, a test designed as a maximal multistage 20 m shuttle run test was implemented [20]. This test was performed individually; participants were asked to run back and forth on a 20 m course and touch the corresponding line while a prerecorded tape emitted a sound signal [21]. As they run, the sound signals’ frequency increases by 0.5 km each minute [21]. The test was finished when participants stopped following the pace suggested by the prerecorded tape. At that moment, the last stage number is announced by a colleague and then used to predict VO2 max [21]. The longer the participants run, the higher the rate of the estimated oxygen uptake [20]. Regarding the reliability of the test, there is a consensus that using VO2max as a measurement criterion for aerobic fitness is high and acceptable [20].

##### Neuromuscular Fitness

In this study, neuromuscular fitness was the third component assessed. According to the literature, muscular strength, muscular endurance, and flexibility are considered critical dimensions of health-related fitness, improving the overall quality of life [20]. There is sustained evidence that enhanced musculoskeletal fitness in children and adolescents is associated with improved overall health status and a reduced risk for chronic disease. Several tests were used in this component to assess overall neuromuscular fitness. Particularly, the dimensions considered and matching tests were: (i) upper-body muscular fitness (Push-ups and Sit-ups tests); (ii) lower-body flexibility (Sit and Reach test); (iii) lower-body muscular fitness (horizontal jump test), and (iv) speed (sprint at 20 m). The tests were implemented during PE classes, with specific and structured instructions and a predefined cadence, and were executed individually by each participant.

### 2.3. Statistical Analysis

The software IBM SPSS Statistics, version 28.0, was used to conduct descriptive statistical analyses. Additionally, to analyze overall changes between the three moments, repeated measure ANOVAs were developed. The Greenhouse–Geisser adjustment was used to correct sphericity violations. For the ANOVAs, partial eta squared was used to determine the effect size (≥0.01 = small, ≥0.06 = medium, ≥0.14 = large) [22]. All tests were two-tailed, and a *p*-value < 0.05 was considered statistically significant. Bonferroni correction was used for post hoc tests. In addition, to explore differences between age (i.e., early and middle adolescents’ samples) and, subsequently, sex (i.e., girls and boys), mixed model ANOVAs were developed. Considering that the assumptions of parametric tests (e.g., normality, homogeneity) were violated for most of the variables, we followed the strategy of computing both parametric and their equivalent nonparametric tests as suggested by Fife-Schaw [23]. Given that the conclusions drawn from both sets of tests were similar in most cases, we opted for presenting the parametric test results because they are more robust and conservative. Missing data were not interpolated. 

## 3. Results

### 3.1. Overall Changes in Physical Fitness over Time 

As shown in Table 2, a repeated measure ANOVA with a Greenhouse–Geisser correction determined that the mean value of BMI HFZc, WC, VO2max, Sit-ups, Push-ups, Horizontal Jumps, and Sit and Reach differed significantly across the three time points (*F* (1.376, 712.705) = 3.864, *p* < 0.05, η_p_2 = 0.007, *F* (1.267, 186.209) = 13.301, *p* < 0.001, η_p_2 = 0.083, *F* (1.675, 845.629) = 73.620, *p* < 0.001, η_p_2 = 0.127, *F* (1.784, 731.531) = 15.087, *p* < 0.001, η_p_2 = 0.035, *F* (1.643, 699.882) = 8.447, *p* < 0.001, η_p_2 = 0.019, *F* (1.848, 770.637) = 76.767, *p* < 0.001, η_p_2 = 0.155, and *F* (1.634, 715.831) = 896.436, *p* < 0.001, η_p_2 = 0.104, respectively). Conversely, there were no statistically significant differences in the 20 m Run across time (i.e., speed). Post hoc pairwise comparisons using the Bonferroni correction were developed between the three moments: 1st Physical Fitness Assessment (T1), 2nd Physical Fitness Assessment (T2), and 3rd Physical Fitness Assessment (T3). The results will be presented in the following subsection.

#### 3.1.1. T1 vs. T2

Between T1 and T2, we found a statistically significant increase in BMI HFZc (*p* < 0.05) and in WC (*p* < 0.001) and a statistically significant decrease in VO2max (*p* < 0.001). Additionally, a statistically significant increase in Horizontal Jumps (*p* < 0.001) and Sit and Reach (*p* < 0.001) were found. However, no statistically significant difference in upper-body muscular fitness (i.e., Sit-ups and Push-ups) was found. Overall, compared with T1, there seems to be a decline in participants’ body composition and aerobic fitness and an improvement in participants’ neuromuscular fitness (i.e., Horizontal Jumps and Sit and Reach) at T2, the return to in-person classes.

#### 3.1.2. T2 vs. T3

Between T2 and T3, we found a statistically significant decrease in BMI HFZc (*p* < 0.05) and WC (*p* < 0.05) and a statistically significant increase in VO2max (*p* < 0.001), Sit-ups (*p* < 0.001), Push-ups (*p* < 0.001), and Horizontal Jumps (*p* < 0.01). However, no significant differences were found in the lower-body flexibility. These results suggest an improvement in participants’ body composition (i.e., BMI HFZc), aerobic fitness, and neuromuscular fitness from T2 to T3 two months after the return to in-person classes.

#### 3.1.3. T1 vs. T3

Between T1 and T3, we found a statistically significant increase in WC (*p* < 0.01) and a statistically significant decrease in VO2max (*p* < 0.001). On the other hand, there is a statistically significant increase in Sit-ups (*p* < 0.001), Push-ups (*p* < 0.01), Horizontal Jumps (*p* < 0.001), and Sit and Reach (*p* < 0.001). In general, compared to T1, the results showed a decline in body composition (i.e., WC) and aerobic fitness and an improvement in participants’ neuromuscular fitness in T3, two months after the return to in-person classes (see Table 2).

### 3.2. Physical Fitness Changes over Time in Early and Middle Adolescents

Considering the age groups, early and middle adolescents (see Table 3), a mixed model ANOVA with a Greenhouse–Geisser correction showed statistically significant differences in WC, Sit-ups, Push-ups, Horizontal Jumps, and Sit and Reach across the three time points (*F* (1.289, 188.125) = 11.507, *p* < 0.001, η_p_2 = 0.073, *F* (1.791, 732.593) = 4.207, *p* < 0.05, η_p_2 = 0.01, *F* (1.633, 694.171) = 4.335, *p* < 0.05, η_p_2 = 0.01, *F* (1.866, 776.174) = 10.822, *p* < 0.001, η_p_2 = 0.025, and *F* (1.646, 719.336) = 36.725, *p* < 0.001, η_p_2 = 0.078, respectively). Conversely, we found non-significant differences between the age groups in BMI HFZc, VO2max, and 20 m Run across the three-time points [*F* (1.377, 711.718) = 1.280, *p* > 0.05, *F* (1.673, 843.387) = 0.518, *p* > 0.05, and *F* (1.516, 706.474) = 0.447, *p* > 0.05, respectively]. The results are stated in the following:

Regarding body composition, compared to early adolescents, middle adolescents showed an increase in WC between T1 (*M* = 68.92, *SD* = 21.37) and T2 (*M* = 77.24, *SD* = 13.92) and between T1 and T3 (*M* = 75.98, *SD* = 13.20), which indicates a significant decline in the health indicator over time for the older participants. 

Regarding neuromuscular fitness, early adolescents, compared to middle adolescents, showed an increase in Sit-Ups between T1 (*M* = 24.25, *SD* = 16.95) and T2 (*M* = 26.81, *SD* = 18.32) and between T2 and T3 (*M* = 29.46, *SD* = 19.81). In addition, this age group showed an increase in Push-ups between T2 (*M* = 9.20, *SD* = 6.52) and T3 (*M* = 9.84, *SD* = 7.00), whereas the middle adolescents showed an increase in Push-ups between T2 (*M* = 12.06, *SD* = 9.06) and T3 (*M* = 14.48, *SD* = 9.53) and between T1 (*M* = 11.82, *SD* = 28.45) and T3. In terms of Horizontal Jumps, compared to middle adolescents, early adolescents showed an increase between T1 (*M* = 128.95, *SD* = 25.77) and T2 (*M* = 138.84, *SD* = 29.47) and between T2 and T3 (*M* = 141.35, *SD* = 30.17). Finally, for the Sit and Reach test, early adolescents showed an increase between T1 (*M* = 17.25, *SD* = 7.23) and T2 (*M* = 20.32, *SD* = 7.97) and T1 and T3 (*M* = 20.12, *SD* = 7.94). Instead, the middle adolescents showed a decrease between T1 (*M* = 22.05, *SD* = 7.90) and T2 (*M* = 19.90, *SD* = 8.39), and an increase between T2 and T3 (*M* = 22.66, *SD* = 7.96). Generally, results indicate that the early adolescent group seems to have improved neuromuscular fitness over time (e.g., upper-body muscular fitness). In contrast, middle adolescents declined or maintained some of the body composition and neuromuscular fitness parameters over time. For a better visual representation of the physical fitness test mean values regarding age groups, see Figure 2.

Finally, considering data for early and middle adolescents over time, additional differences between boys and girls were developed for each age subgroup (see Appendix A). In fact, when layered by sex, the results of the ANOVA with Greenhouse–Geisser correction showed that, for early adolescents, the only significant differences between sex across time were found in the following neuromuscular fitness parameters: Horizontal Jumps, 20 m Run, and Sit and Reach (*F* (1.882, 617.393) = 8.927, *p* < 0.001, η_p_2 = 0.026, *F* (1.540, 574.400) = 6.488, *p* < 0.01, η_p_2 = 0.017, and *F* (1.556, 541.327) = 8.067, *p* < 0.01, η_p_2 = 0.023). Moreover, boys performed better over time in the Horizontal Jump and 20 m Run tests than girls, while girls revealed a higher performance in Sit and Reach. However, both sex groups improved over time in these indicators (see Appendix A). 

For the middle adolescents, there was a statistically significant difference between girls and boys over time only in the Horizontal Jumps neuromuscular parameter [*F* (1.803, 155.060) = 3.503, *p* < 0.05, η_p_2 = 0.039]. Data show that boys showed an increase between T1 (*M* = 179.54, *SD* = 4.30) and T3 (*M* = 187.68, *SD* = 4.27). Nonetheless, no differences were found over time in girls. In addition, for this age group, no significant differences between boys and girls were found in BMI HFZc, WC, VO2max, Sit-ups, Push-ups, Sit and Reach, and 20 m Run across the three time points: (*F* (1.423, 136.630) = 0.452, *p* > 0.05, *F* (1.103, 61.744) = 0.230, *p* > 0.05, *F* (1.613, 141.922) = 0.074, *p* > 0.05, *F* (1.768, 139.698) = 0.798, *p* > 0.05, *F* (1.774, 143.722) = 1.342, *p* > 0.05, *F* (1.405, 127.838) = 0.702, *p* > 0.05, and *F* (1.866, 162.336) = 0.356, *p* > 0.05, respectively).

## 4. Discussion

The present study is one of the few longitudinal studies that focused on the potential impact of the COVID-19 lockdown on adolescents’ physical fitness. For that purpose, several physical fitness parameters were assessed over time, particularly before (T1) and after the first lockdown in Portugal (T2 and T3). Furthermore, to deepen our understanding of the COVID-19 lockdown impact, the current study also explored differences over time between two age groups and the variable sex. 

Overall, current findings suggest that participants’ body composition (i.e., WC) and aerobic fitness (i.e., VO2max) deteriorated during the period of online classes but improved two months after the return to in-person classes. However, in terms of the BMI HFZc, although a significant increase was observed, the magnitude of the observed effect was relatively small; therefore, this finding should be considered with caution. In addition, the pattern seen in both body composition and aerobic fitness was not observed in neuromuscular fitness (i.e., Horizontal Jumps and Sit and Reach). In fact, contrary to other studies (e.g., Wahl-Alexander et al. [15]), current data shows that the latter variables improved (e.g., Horizontal Jumps) or remained constant (e.g., Push-ups) over time. These results are consistent with those by Basterfield et al. [11], indicating that during the lockdown children’s aerobic fitness (i.e., VO2max) was more susceptible to change than neuromuscular fitness parameters (e.g., Horizontal Jumps). As suggested by the same authors, a decline in body composition and aerobic fitness may reflect possible changes in children’s physical behaviors during the pandemic lockdown [11]. For example, some of the measures imposed in Portugal to mitigate the pandemic’s effects, including school closure (i.e., for more than three months [24]) and other social distancing norms (e.g., the mandatory use of social masks), are likely to have limited students’ engagement in school and out-of-school activities (e.g., collective and outdoor sports) and interaction with friends and classmates [5,6,7]. In fact, as the literature reports, adolescents enclosed at home were prone to engage in sedentary activities such as screen time [25,26]. All this considered, the Portuguese measures to mitigate the pandemic’s effects may have influenced adolescents’ physical behaviors at T2. Moreover, the return to school and in-person classes (i.e., the period between T2 and T3) helped students recover their daily routines; for example, they gradually returned to engage in social and physical activities interrupted by the pandemic lockdown. This return to “normality” may have helped adolescents recover their physical activity levels and improve their physical fitness parameters.

Focusing on the analysis of the age groups, findings indicate significant differences between early and middle adolescents in WC, Sit-ups, Push-ups, Horizontal Jumps, and Sit and Reach, although with a small to medium-size effect of age (i.e., from 0.01 to 0.13). Specifically, for early adolescents, contrary to what was expected, neuromuscular fitness (i.e., Sit-ups, Push-ups, Horizontal Jumps, and Sit and Reach) seem to have improved after the period of online classes (from T1 to T2) and two months after the return to in-person classes (T3). In fact, in line with Rossi et al. [27], several facilitating factors (i.e., individual, and contextual factors) can shed some light on these results. For instance, considering individual factors, age seemed to be an important variable influencing the physical activity levels, especially among early adolescents who had frequent PE online classes and followed structured physical routines at home [27]. Furthermore, young students were more likely to participate in free play and unstructured physical activity during leisure time (e.g., biking) than their older peers [28]. Regarding the contextual factors, previous studies showed that those living in rural environments were more prompt to engage in physical activities than those living in an urban environment [29]. In fact, rural environments—such as the one we investigated in this study—are expected to allow more opportunities for physical activity and free play (i.e., outdoor activities) while following the social distancing measures than urban environments [27]. Moreover, we anticipate that these outdoor activities (e.g., riding on bikes) were more appealing to younger age groups (e.g., early adolescents). All the factors above help us explain results indicating an improvement in several neuromuscular fitness parameters for the younger age group (i.e., early adolescents). 

However, considering the results for early adolescents, we analyzed the differences between girls and boys. Data indicated that girls and boys showed significant differences over time despite the small effect size (i.e., from 0.01 to 0.05). However, boys and girls showed an improvement in Horizontal Jumps, 20 m Run, and Sit and Reach tests at each time point. Interestingly, these results are consistent with Marta et al. [30], who investigated early adolescents and concluded that boys are likely to perform better at speed and muscular strength tests than girls and that girls are more prone than boys to show high levels of flexibility. In addition, a recent systematic review indicated that being a boy can be considered a facilitating factor for maintaining high levels of physical activity [27], which is consistent with the current results found for boys over time. 

The trend was slightly different for middle adolescents compared to the early adolescents’ subgroup. Particularly, for most neuromuscular fitness parameters (i.e., Sit-ups, Push-ups, and Horizontal Jumps), the middle adolescents showed similar scores from T1 to T2. Whereas for WC and Sit and Reach, the middle adolescents seemed to have deteriorated during the online classes period (T1 to T2) and improved two months after the return to in-person classes (T2 to T3). Therefore, this age group seems to have been more affected by the lockdown and the online classes period than the early adolescents. The latter result is concerning, particularly for WC, which is seen as a cardiovascular risk factor and an indicator of visceral obesity, a central factor for vascular impairment in adolescents [31]. Regarding this anthropometric indicator, previous findings support the result that there were losses in adolescents after the COVID-19 lockdown (e.g., Wahl-Alexander and Camic, Rúa-Alonso et al., and Tsoukos and Bogdanis [12,13,15]). For instance, a recent cross-sectional study with two samples of children and adolescents, assessed before and during the COVID-19 pandemic, revealed higher increases in BMI and WC over this period [12]. Regarding neuromuscular fitness, adolescents, particularly boys, tend to show better performance as they grow [32]. However, as previously stated, older adolescents in the current study showed less improvement in neuromuscular fitness parameters over time than their younger counterparts. Furthermore, no differences were found between girls and boys, except for the horizontal jump test, where boys seem to have improved over time. Altogether, we can conclude that for this cohort the physical activity during the COVID-19 lockdown was insufficient for maintaining a proper growth of the neuromuscular parameters. 

In sum, the results of this study strengthen the shared idea that the COVID-19 mandatory lockdown and the restrictions implemented worldwide, particularly in Portugal (e.g., school closure, online classes, and social distancing) may have had a relevant effect on the health habits of the youth (e.g., eating behaviors and physical activity [5]), (i) influencing the increase of adiposity in adolescents, particularly in the older age group, and (ii) impacting the older adolescents’ physical and muscular growth. These results are consistent with other preliminary reports [12,13,15]. 

The present study has some limitations that should be considered. We must be careful in generalizing these results since our sample is restricted to two rural public schools in the north of Portugal rather than being geographically distributed. In addition, we did not collect qualitative data (e.g., students’ level of habitual physical activity) or self-report data that could have helped better understand the results (e.g., declarative knowledge on the value of physical exercise for health and self-efficacy to physical exercise). There is also a considerable amount of missing data regarding the current quantitative data. In this regard, we can hypothesize several reasons, mainly participants missing the PE class, transference to another school, or, in the case of WC, students refusing to do the measurements. Furthermore, the schools underwent a period of reorganization of the school management right after the first lockdown in Portugal (e.g., creating an online system to deliver classes and rescheduling students’ and teachers’ classes). Finally, due to the lack of a control group, we cannot state that the changes in physical fitness parameters are directly associated with the COVID-19 lockdown and its restrictions. In fact, some changes could be attributed to the expected development growth occurring during adolescence, for instance, growth in height, weight, strength, and muscular endurance. Note that this growth can be distinct among girls and boys [17,32]. All considered these results must be interpreted cautiously. 

On the other hand, this study has several strengths that should be considered. This study provides a baseline before the COVID-19 pandemic in Portugal and two measures during the pandemic, making it possible to analyze longitudinal patterns and trends due to extended absence from school. By studying these variables over time, we are contributing to the current body of research that aims to highlight the importance of these indicators for adolescents’ future health since adolescence is a critical transition period between childhood and adulthood [14,17]. 

In addition, we are contributing to understanding how the COVID-19 pandemic could have impacted these physical fitness indicators (e.g., body composition). Furthermore, we hope these findings could encourage educators’ work in their practices to promote physical activity. This is an important ongoing task because recent World Health Organization data [33] reports that 81% of adolescents aged 11 to 17 years do not do enough physical activity, making sedentarism a real and serious issue. 

## 5. Conclusions

In this study, adolescents’ physical fitness, mainly body composition and aerobic fitness, appeared to have changed after the lockdown, leading us to speculate that the first COVID-19 lockdown could have impacted adolescents’ physical fitness, particularly for older adolescents. These findings reinforce the need to continue promoting and evaluating adolescents’ health behaviors (e.g., physical activity), specifically in the school context. 

Furthermore, this study sheds light on some of the possible implications of the COVID-19 pandemic on adolescents’ health. These should be considered for designing future interventions to diminish pandemic restrictions’ negative effects on health-related fitness. Considering this, it is imperative to continue studying these variables in adolescent samples.

## Figures and Tables

**Figure 1 ijerph-20-03422-f001:**
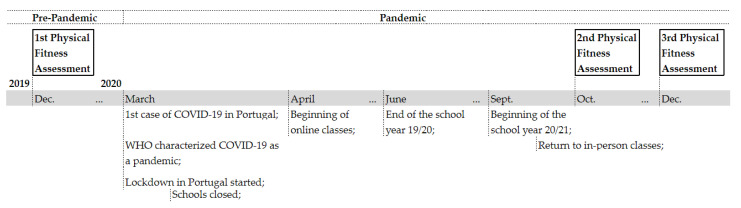
Physical fitness assessment timeline and COVID-19 regulations.

**Figure 2 ijerph-20-03422-f002:**
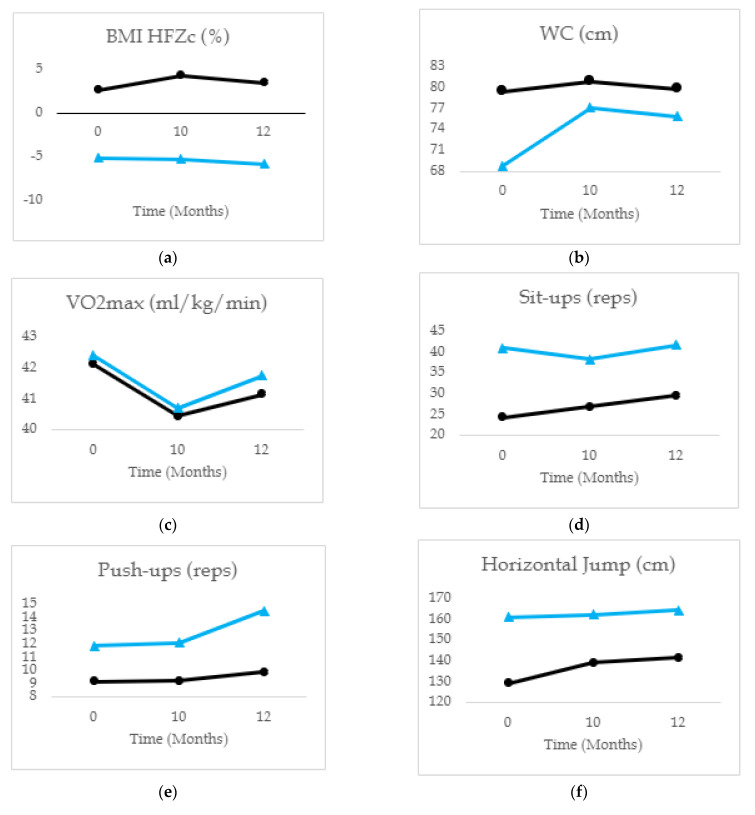
Comparison of the physical fitness test´s mean values in the three moments and between the subgroups of early and middle adolescents; (**a**) comparison of BMI HFZc mean values in the three moments and between the subgroups (early and middle adolescents); (**b**) comparison of WC mean values in the three moments and between the subgroups (early and middle adolescents); (**c**) comparison of VO2max mean values in the three moments and between the subgroups (early and middle adolescents); (**d**) comparison of Sit-Ups mean values in the three moments and between the subgroups (early and middle adolescents); (**e**) comparison of Push-Ups mean values in the three moments and between the subgroups (early and middle adolescents); (**f**) comparison of Horizontal Jumps mean values in the three moments and between the subgroups (early and middle adolescents); (**g**) comparison of 20 m Run mean values in the three moments and between the subgroups (early and middle adolescents); (**h**) comparison of Sit and Reach mean values in the three moments and between the subgroups (early and middle adolescents).

**Table 1 ijerph-20-03422-t001:** Sample descriptive statistics.

Participants		T1	T2	T3
		*n (%)*	*n (%)*	*n (%)*
Sex				
	Girls	305 (47.7)	273 (47.5)	280 (47.5)
	Boys	335 (52.3)	302 (52.5)	310 (52.5)
Subgroup				
	Early Adolescents	497 (77.7)	360 (62.6)	338 (57.3)
	Middle Adolescents	143 (22.3)	215 (37.4)	252 (42.7)

**Table 2 ijerph-20-03422-t002:** Repeated measure ANOVA and Bonferroni Pairwise Comparisons for the total sample.

	Pre-Pandemic	Pandemic					Bonferroni Pairwise Comparisons		
	T1 (December 2019) *M (SD)*	T2 (October 2020)*M (SD)*	T3 (December 2020) *M (SD)*						
						η_p_2	T1 vs. T2	T2 vs. T3	T1 vs. T3
**Body Composition**									
BMI HFZc (%) (*n* = 519)	1.14 (21.59)	2.45 (21.90)	1.68 (21.31)	*F* (1.376, 712.705) 3.864 *		0.007	*	*	ns
WC (cm) (*n* = 148)	75.39 (16.79)	79.53 (11.97)	78.36 (11.55)	*F* (1.267, 186.209) 13.301 ***		0.083	***	*	**
**Aerobic Fitness**									
VO2max (ml/kg/min) (*n* = 506)	42.15 (5.45)	40.48 (5.19)	41.24 (5.53)	*F* (1.675, 845.629) 73.620 ***		0.127	***	***	***
**Neuromuscular Fitness**									
Sit-ups (*n* = 411)	27.55 (19.22)	29.04 (19.97)	31.85 (20.95)	*F* (1.784, 731.531) 15.087 ***		0.035	ns	***	***
Push-ups (*n* = 427)	9.65 (7.70)	9.75 (7.17)	10.74 (7.77)	*F* (1.643, 699.882) 8.447 ***		0.019	ns	***	**
Horizontal Jump (*n* = 418)	135.64 (31.32)	143.73 (32.92)	146.18 (33.84)	*F* (1.848, 770.637) 76.767 ***		0.155	***	**	***
20 m Run (*n* = 468)	4.34 (0.62)	4.31 (0.57)	4.30 (0.58)	*F* (1.516, 707.765) 1.722 (ns)		0.004	ns	ns	ns
Sit and Reach (*n* = 439)	18.23 (7.61)	20.24 (8.05)	20.64 (8.00)	*F* (1.634, 715.831) 896.436 ***		0.104	***	ns	***

* *p* < 0.05, ** *p* < 0.01, *** *p* < 0.001; ns—non-significant value; BMI HFZc—Body Mass Index Health Fitness Zone Continuum; WC—waist circumference; VO2max—maximal oxygen uptake.

**Table 3 ijerph-20-03422-t003:** Mixed model ANOVA and Bonferroni pairwise comparisons of early and middle adolescents’ subgroups.

		Pre-Pandemic	Pandemic				Bonferroni Pairwise Comparisons		
		T1 (Dec 2019) *M (SD)*	T2 (Oct 2020) *M (SD)*	T3 (Dec 2020) *M (SD)*					
						η_p_2	T1 vs. T2	T2 vs. T3	T1 vs. T3
**Body Composition**									
BMI HFZc (%) (*n* = 519)	**Early** **Adolescents** (*n* = 421)	2.61 (21.63)	4.26 (21.53)	3.43 (20.43)	*F* (1.377, 711.718) = 1.280 (ns)	0.002	*	*	ns
	**Middle** **Adolescents** (*n* = 98)	−5.16 (20.35)	−5.32 (21.88)	−5.84 (22.11)			ns	ns	ns
WC (cm) (*n* = 148)	**Early** **Adolescents** (*n* = 90)	79.56 (11.33)	81.01 (10.35)	79.90 (10.13)	*F* (1.289, 188.125) = 11.507 ***	0.073	ns	ns	ns
	**Middle** **Adolescents** (*n* = 58)	68.92 (21.37)	77.24 (13.92)	75.98 (13.20)			***	ns	***
**Aerobic Fitness**									
VO2max (ml/kg/min) (*n* = 506)	**Early** **Adolescents** (*n* = 416)	42.10 (5.00)	40.43 (4.52)	41.14 (4.91)	*F* (1.673, 843.387) = 0.518 (ns)	0.001	***	***	***
	**Middle** **Adolescents** (*n* = 90)	42.39 (7.18)	40.69 (7.57)	41.73 (7.81)			***	***	ns
**Neuromuscular Fitness**									
Sit-ups (*n* = 411)	**Early** **Adolescents** (*n* = 330)	24.25 (16.95)	26.81 (18.32)	29.46 (19.81)	*F* (1.791, 732.593) = 4.207 *	0.010	*	***	***
	**Middle** **Adolescents** (*n* = 81)	40.96 (22.00)	38.16 (23.63)	41.60 (22.72)			ns	ns	ns
Push-ups (*n* = 427)	**Early** **Adolescents** (*n* = 344)	9.13 (7.42)	9.20 (6.52)	9.84 (7.00)	*F* (1.633, 694.171) = 4.335 *	0.010	ns	*	ns
	**Middle** **Adolescents** (*n* = 83)	11.82 (8.45)	12.06 (9.06)	14.48 (9.53)			ns	***	***
Horizontal Jump (*n* = 418)	**Early** **Adolescents** (*n* = 330)	128.95 (25.77)	138.84 (29.47)	141.35 (30.17)	*F* (1.866, 776.174) = 10.822 ***	0.025	***	**	***
	**Middle** **Adolescents** (*n* = 88)	160.73 (37.20)	162.08 (38.45)	164.28 (40.35)			ns	ns	ns
20 m Run (*n* = 468)	**Early** **Adolescents** (*n* = 375)	4.42 (0.59)	4.40 (0.55)	4.39 (0.56)	*F* (1.516, 706.474) = 0.447 (ns)	0.001	ns	ns	ns
	**Middle** **Adolescents** (*n* = 93)	4.01 (0.62)	3.95 (0.51)	3.93 (0.49)			ns	ns	ns
Sit and Reach (*n* = 439)	**Early** **Adolescents** (*n* = 350)	17.25 (7.23)	20.32 (7.97)	20.12 (7.94)	*F* (1.646, 719.336) = 36.725 ***	0.078	***	ns	***
	**Middle** **Adolescents** (*n* = 89)	22.05 (7.90)	19.90 (8.39)	22.66 (7.96)			***	***	ns

* *p* < 0.05, ** *p* < 0.01, *** *p* < 0.001; ns—non-significant value; BMI HFZc—Body Mass Index Health Fitness Zone Continuum; WC—waist circumference; VO2max—maximal oxygen uptake. The subgroup of early adolescents includes participants between 10 and 13 years of age; the subgroup of middle adolescents includes participants between 14 and 18 years of age.

## Data Availability

Data is available from the first author upon reasonable request.

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
