# Peer review of "Changes in Physical Fitness Parameters in a Portuguese Sample of Adolescents during the COVID-19 Pandemic: A One-Year Longitudinal Study"

_ijerph, 2023, doi:10.3390/ijerph20043422_

Round 1

Reviewer 1 Report

   The findings of this study, which examined longitudinal changes in adolescents’ physical fitness parameters before the COVID-19 pandemic, when the school reopened and at two months after the COVID-19 lockdown, are interesting. However, due to the study population was a developmental population and that the T1-T2 and T2-T3 time periods were very different, it was deemed necessary to re-analyses, add discussion and revise the figures, as listed below.

      The period from T1 to T2 is 10 months, whereas the period from T2 to T3 is only 2 months. In particular, it should be noted that for the subject of Early Adolescents, the change from T1 to T2 has additional developmental effects in addition to the effects of lockdown. Furthermore, it should be noted that these subjects are a developmentally crossed population. In other words, Early Adolescents subjects are likely to have greater height and weight gain in females than in males, whereas Middle Adolescents subjects are likely to have greater height and weight gain in males than in females.

   For the reasons stated above, it is considered that the analysis should be conducted separately for males and females.

   Whether or not there is a difference in the longitudinal change between males and females as a result of the re-analysis, it should be mentioned in the discussion that there is an additional effect of developmental growth and development on the change from T1 to T2 period.

   In addition, the intervals between T1, T2 and T3 on the X-axis of all graphs should vary in width to match the physical time interval.

   As the period between T1 and T3 is exactly one year, the quality of the study would be improved if the longitudinal changes of the study population and the same age group in the same region during the year 2018-2019 were compared and discussed with the results of this study, if data are available.

Reviewer 2 Report

Comments and Suggestions for Authors

Thank you for the opportunity to review this manuscript. The purpose of the authors was to comprehend the impact of early COVID-19 pandemic restrictions on ad-14 adolescents physical fitness parameters during two school years. Despite the importance of this topic, and also the efforts of the authors to present a clear idea about the study performed, I have a few concerns about this paper that I request that the authors address before the manuscript is ready for publication

 Abstract

Acronyms presented should be explained before their first use.

L24: “These findings suggest that the 24 COVID-19 lockdown may have impacted adolescents’ physical fitness”. Please, try to move beyond the obvious. How can these findings be used in a practical context? Or, what are the implications of these findings?

Keywords: The authors should try to use different words from those used in the title

Introduction

L31 – 33: Please, provide a reference to support the sentence

L78: likely to help educators in their practices: please be more specific

There is a massive information about the topic of the present study in the scientific literature. The authors must be clear about what is the novelty of this study. Further, the authors highlighted the design of the study as an important aspect for justifying the relevance of the study, however, it could be good to present additional information about the role of “the time” (time as a predictor) for changes in physical fitness parameters in the introduction.

Methods

-        How was sociodemographic information obtained?

-        Were normality and homogeneity of the variances previously tested?

-    Do the authors consider the effect of sex? Why statistical analysis was not stratified by sex?

-        Please, consider the possibility to include a between-groups comparison (early and middle age) over time.

Results

-        Table 2: The authors could try to reduce the row size, using only T1, T2 and T3. Please, include a legend to clarify the meaning of acronyms.

-        Figure 2: should be improved

-        The authors need to discuss the results using the effect size information.

-        The authors presented information considering different age groups. What is the theoretical rationale for this approach? It should be clarified in the introduction.

Discussion

-        In the first paragraph of the discussion, the authors presented information about the time of the data collection. Since this information was presented in the methods section, it is redundant and can be removed from the discussion.

-        The main findings of the authors needs to be followed by the magnitude of the effect of the change. Please, be transparent about it.

-        The authors also found that some physical fitness indicators improved two months after the return of the in-person classes. What do explain these findings?

-        Discussion is poor. The authors only repeat the main results and present some previous findings. The main question to be considered in the discussion is: why? Or, what does explain these findings? How the changes in specific physical fitness components are related to health outcomes? That is, what are the outcomes of the changes in aerobic and neuromuscular fitness?

-        What variables were not measured but could have played a relevant role in  the results?

-        The authors mentioned that the sample is from two rural schools in Portugal and also that they have  information about the family income status (mentioned at the methods section). How do these factors affect the results?

-        L392: “In addition, we are contributing to understanding how the COVID-19 pandemic 392 could have impacted these indicators” – Please, specify the indicators.

Reviewer 3 Report

REVIEW OF MANUSCRIPT: ijerph-2160546

This is a novel and well written exploration of the impacts of COVID on adolescent fitness. There are just some minor changes that could help strengthen the study.

ABSTRACT

This is a great overview of the study however, I would suggest including where the study took place for clarity as lockdown periods would have been different globally.

INTRODUCTION

This is a comprehensive and clear introduction to the topic and study in question. I only have one suggestion:

P1, L37: What are the changes to our lifestyle you are referring to? More sedentary, diet, cost of living? All?

METHODS

This is a good overview but some further clarity on data collection methods would allow for reproducibility under similar protocols.

P3, L106: How was socio-economic position defined?

P4, L139: Were researchers involved at all? Did they oversee or was this left entirely to the school’s discretion? Worth noting for reliability.

P4, L143: Who collected these?

P4, L162: Where was this carried out? In schools? Under what supervision?

RESULTS

Clear and concise.

DISCUSSION

This is a good description of findings but presents no context or link to previous literature/studies around why these trends have been observed. This section would benefit from a similar approach to the introduction where previous literature is referred to and highlighted as similar/providing context.

It would be worth discussing what school closures looked like in Portugal. In the UK, this meant a switch to online learning, pressures on parents to teach and guide as well as less time with friends and playing. There has been an argument for an against whether this improved activity levels. It would be good to hear about this from a Portuguese perspective with the inclusion of anthropological measures which is novel.

CONCLUSIONS

Could be strengthened by changes to the discussion.

Round 2
